# EFFICIENT ON-POLICY REINFORCEMENT LEARNING VIA EXPLORATION OF SPARSE PARAMETER SPACE

## ABSTRACT

Policy-gradient methods such as Proximal Policy Optimization (PPO) are typically updated along a single stochastic gradient direction, leaving the rich local structure of the parameter space unexplored. Prior work has shown that the surrogate gradient is often poorly correlated with the true reward landscape. Building on this insight, we visualize the parameter space spanned by policy checkpoints within an iteration and reveal that higher-performing solutions often lie in nearby unexplored regions. To exploit this opportunity, we introduce ExploRLer, a pluggable pipeline that seamlessly integrates with on-policy algorithms such as PPO and TRPO, systematically probing the unexplored neighborhoods of surrogate on-policy gradient updates. Without increasing the number of gradient updates, ExploRLer achieves significant improvements over the baselines in complex continuous control environments. Our results demonstrate that iteration-level exploration provides a practical and effective way to strengthen on-policy reinforcement learning and offer a fresh perspective on the limitations of the surrogate objective.

## 1 INTRODUCTION

On-policy reinforcement learning (RL) methods such as TRPO (Schulman et al., 2015a) and PPO (Schulman et al., 2017) have become foundational tools for both classic benchmarks (e.g. MuJoCo locomotion, DM Control Suite) and modern applications including large language model alignment and fine-tuning (Raffin et al., 2021; Ouyang et al., 2022; Rafailov et al., 2023; Shao et al., 2024). Their appeal lies in theoretically grounded, low-variance updates that maintain a trust region around the current policy, yielding stable learning even in high-dimensional action spaces.

However, despite their guarantees, these methods often converge slowly or stall in practice. A key culprit is the high variance of on-policy gradient estimates: although fresh trajectories ensure unbiasedness, cumulative noise in long-horizon rewards causes gradient estimates to scatter, sometimes leading to uncontrolled updates (Greensmith et al., 2004). Trust-region surrogates bound likelihood ratios to curb this variance, but Ilyas *et al.* show that the surrogate objective can be poorly aligned with the true reward landscape, explaining why PPO may still follow suboptimal search directions (Ilyas et al., 2020).

A typical on-policy learning method maintains a rollout buffer, which is filled with fresh data collected by the current policy at the start of each iteration. During each iteration, the algorithm performs multiple epochs over the rollout buffer. In each epoch, it samples a mini-batch of data to estimate the gradient and update the policy parameters. Building on this training process, traditional approaches such as augmenting each gradient step with random search corrections, sample large perturbation batches during mini-batch updates to statistically refine the update direction. While these mini-batch-level methods can improve the policy update direction, they also impose a heavy computational and data burden, often requiring hundreds of parallel rollouts per update.

In this work, we take a different view: Instead of correcting each mini-batch, we systematically probe the richer *local parameter space* around on-policy RL's iterations. Building on Empty-Space Search Algorithm (ESA) (Zhang et al., 2025b), our method, ExploRLer, works at the *iteration level*: 1. Anchor: After every fixed number of RL iterations, collect the last-step checkpoint from each iteration as anchors. 2. Explore: Apply an empty-space operator to generate candidate policies in the surrounding parameter region. 3. Evaluate & Resume: Evaluate these candidates by online interactions and restart training from the best one. By shifting exploration to the iteration level,

ExploRLer requires *zero additional gradient computations* per mini-batch, yet it uncovers high-reward directions missed by on-policy RL alone.

We summarize our contributions as the following:

- **ExploRLer Pipeline**. We introduce a novel *parameter spacE eXPloration pipeLine for On-policy Reinforcement Learning Efficiency impRovement* that augments on-policy RL with targeted empty-space search at the iteration granularity, correcting surrogate-gradient bias without extra gradient overhead.

- **Training Granularity**. We formalize batch-level, epoch-level, and iteration-level views of on-policy training, and empirically demonstrate that iteration-level exploration achieves both computational efficiency and search effectiveness. Our granularity framework provides a guideline for future algorithm design on gradient correction.

- **Comprehensive Evaluations**. We compare our method against multiple state-of-the-art baselines and alternative design choices across a variety of locomotion environments. We also provide preliminary evidence of applicability to the off-policy domain and investigate strategies for improving sample efficiency.

Through extensive benchmarks on MuJoCo, Box2D, and Classic control tasks, ExploRLer consistently improves final returns and convergence speed over on-policy RL algorithms, offering a practical route to more robust on-policy reinforcement learning.

## 2 RELATED WORK

**On-Policy RL**    On-policy RL is valued for its clear theoretical guarantees and simple architecture, which rely on consistency between the data distribution and the policy distribution. Natural Policy Gradient (Kakade, 2001) and REINFORCE (Williams, 1992) established the foundational stochastic gradient estimation framework. (Schulman et al., 2015a) introduced TRPO, which enforces monotonic improvement via a KL-divergence–based trust region, ensuring both stable convergence and controlled step sizes. PPO (Schulman et al., 2017) further simplified TRPO with a clipped surrogate objective, enabling efficient first-order updates and widespread adoption. Subsequent variants – dual-clip PPO (Ye et al., 2020), which adds a lower clip bound for negative-advantage samples, and soft-clip PPO (Chen et al., 2023), which permits a larger trust region – address extreme importance ratios and larger policy shifts. Nevertheless, on-policy methods can still exhibit instability and low sample efficiency in practice. Various heuristics have been proposed to stabilize training (Huang et al., 2022; Engstrom et al., 2020), but these add complexity without offering an unified design principle.

**Parameter Optimization**    To further reduce gradient variance and improve stability, several zero-order and gradient-correction strategies have been developed. Mania et al. (2018) propose a random search method that achieves unbiased gradient estimates without backpropagation. Maheswaranathan et al. (2019) refine this approach by restricting perturbations to a promising subspace to approximate true policy gradients under a surrogate objective, while Sener & Koltun (2020) argue that objective functions lie on a low-dimensional manifold and introduce an algorithm to learn and search within this latent space. Gao & Sener (2022) further establish a linear relationship between gradient-estimation mean squared error and convergence speed on theory. Rahn et al. (2023) examine noisy neighbors in the return landscape and improve stability by rejecting updates with low post-update CVaR. Although effective, these methods require large sample batches at each update, leading to high computational, memory, and data demands.

**Value-Network Perturbation and Empty-Space Search**    Marchesini & Amato (2023) show that on-policy value functions often become trapped in local optima, which impairs accurate return estimation. To address this, they periodically perturb the value network and select the best-performing variant from the resulting ensemble. Although this strategy can escape local traps, it still relies on random exploration around a single point, resulting in a similar output to Mania et al. (2018). By contrast, ESA (Zhang et al., 2025b) probes the broader parameter space defined by multiple checkpoints. It identifies "empty spaces" and generates new candidates in promising "empty spaces". Prior work (Zhang et al., 2025b;a) demonstrates that ESA can reveal high-performing solutions missed by conventional searching strategies.

**The Gap and Our Contribution**  Despite these advances, no existing method systematically probes the local parameter space at training iterations without introducing per-update overhead. Our ExploRLer pipeline fills this gap by integrating ESA into the on-policy training loop, enabling efficient, bias-corrected exploration of the parameter neighborhood with zero additional gradient computations.

# 3 PRELIMINARY

## 3.1 ON-POLICY REINFORCEMENT LEARNING

Markov Decision Process (MDP) is the basis of RL which consists of the following six-element tuple: $(\rho_0, \mathcal{S}, \mathcal{A}, R, T, \gamma)$, where $\rho_0$ is the initial state distribution; $\mathcal{S}$ is the state space; $\mathcal{A}$ is the action space; $R : \mathcal{S} \times \mathcal{A} \to \mathbb{R}$ is the reward function; $T : \mathcal{S} \times \mathcal{A} \to \mathcal{S}$ is the transition function; $\gamma \in [0, 1)$ is the discount factor. Denote a policy $\pi_\theta(s|a)$ parameterized by $\theta$ that outputs the probability distribution of actions given a state. The objective of on-policy RL is to find $\theta$ that maximizes the average expected discounted return:

$$J(\theta) = E_{\tau \sim \pi_\theta} \left[ \sum_{t=0}^{\infty} \gamma^t R(s_t, a_t) \right] \tag{1}$$

where $\tau = (s_0, a_0, s_1, a_1, \dots)$ denotes a trajectory induced by following $\pi_\theta$ in the environment. The policy gradient theorem Sutton et al. (1999) shows that the gradient of $J(\theta)$ can be written as

$$\nabla_\theta J(\theta) = E_{s \sim d^{\pi_\theta}, a \sim \pi_\theta} \left[ \nabla_\theta log \pi_\theta(a|s) A^{\pi_\theta}(s, a) \right] \tag{2}$$

where $A^{\pi_\theta}(s, a) = Q^{\pi_\theta}(s, a) - V^{\pi_\theta}(s)$ is the advantage function. Furthermore, $A^{\pi_\theta}(s, a)$ is usually estimated by GAE Schulman et al. (2015b) in a per time step manner: $A_t = \sum_{l=0}^{\infty} (\gamma\lambda)^l (r_t + \gamma V(s_{t+1}) - V(s_t))$. However, directly optimizing the policy using Eq. ( 2) is prone to instability due to large policy updates. TRPO addresses this by introducing an approximated learning objective to constrain the updates:

$$L(\theta) = E_{s, a \sim \pi_\theta} \left[ \frac{\pi_\theta(a|s)}{\pi_{\theta_{old}}(a|s)} A^{\pi_\theta}(s, a) \right] \tag{3}$$

PPO further simplifies Eq. ( 3) by clipping the advantage by $1 \pm \epsilon$ and re-expressed as

$$L(\theta) = E_t(\min(r_t(\theta)A_t, \text{clip}(r_t(\theta), 1 - \epsilon, 1 + \epsilon)A_t) \tag{4}$$

where $t$ is the time step, $r_t(\theta) = \frac{\pi_\theta(a_t|s_t)}{\pi_{\theta_{old}}(a_t|s_t)}$. $L(\theta)$ is a surrogate objective of $-J(\theta)$ in Eq. (1).

## 3.2 EMPTY-SPACE SEARCH

**Empty Spaces**  Empty spaces are contiguous regions that contain no observed samples but may hold novel and potentially high-value data. Due to the curse of dimensionality, real-world datasets occupy only a vanishing fraction of the total volume, leaving vast "voids" that standard optimization or sampling methods seldom visit. Identifying and exploiting these voids can reveal out-of-distribution samples that outperform existing solutions.

**Heuristic Search Principle**  Zhang et al. (2025b) proposed an agent-based heuristic method to identify empty spaces. It iteratively pushes a set of particles away from known data until each particle stabilizes in a sparse region. The motion is driven by a Lennard–Jones–style potential for its simplicity and efficiency:

$$\phi(r) = 4\epsilon \left[ \left( \frac{\sigma}{r} \right)^{12} - \left( \frac{\sigma}{r} \right)^6 \right] \tag{5}$$

where $r$ is the distance from the particle to its nearest neighbor; $\sigma$ is the effective diameter that determines the effective size of attractive force and repulsive force; $\epsilon$ is the depth of the potential well that determines the maximal attractive force. The force to guide the particle moving direction is calculated by

$$\vec{F}(r) = \nabla_r \phi(r) = 24\epsilon\sigma \left[ 2 \left( \frac{\sigma}{r} \right)^{13} - \left( \frac{\sigma}{r} \right)^7 \right] \vec{u} \tag{6}$$

where $\vec{u}$ is the unit vector pointing from the neighbor to the particle. $\vec{F}(r)$ repels the particles if $r < \sigma$ and otherwise attracts back.

**Empty-Space Search Algorithm** Empty-space search algorithm (ESA) employs the aforementioned heuristic method to identify empty spaces. It initializes a set of random particles in the data space. The particles move a fixed step size towards the forcing direction for every step. The forcing direction calculation optionally integrates a momentum mechanism that incorporates an exponentially weighted average of past directions to smooth the search trajectory and encourage broader exploration. After stabilization, the particles become centers of empty spaces. ESA periodically releases new data from particle trajectories.

This concise background covers the key concepts behind ExploRLer: on-policy gradient updates and parameter-space exploration (ESA).

## 4 METHOD

We introduce *ExploRLer*, a pipeline that augments on-policy RL with targeted empty-space exploration during training iterations. It alternates between standard on-policy updates and zero-order candidate proposals from ESA. This section first summarizes the challenges in on-policy gradient estimation, then details our adaptation of ESA and the full pipeline.

### 4.1 MOTIVATION: SURROGATE-GRADIENT DRIFT

Ilyas et al. (2020) highlighted three major issues with on-policy gradient estimation: (i) the estimated policy gradient $\nabla_\theta J(\theta)$ exhibits extremely high variance across batches; achieving consistent estimates requires batch size $> 10^6$, far above the typical $\sim 10^3$, (ii) $\nabla_\theta J(\theta)$ is poorly aligned with the true gradient $\nabla_\theta J^*(\theta)$ obtained from extensive online rollouts: matching the true direction again demands $> 10^6$ state–action samples, whereas estimates from $\sim 10^3$ samples are nearly orthogonal, (iii) visualization of the reward landscape shows that PPO's and TRPO's surrogate objectives deviate increasingly from the real reward surface as training progresses.

To further illustrate this, we visualize the distribution of policy values in the local parameter space around a PPO iteration in Figure 1. The results reveal that although checkpoints generally move toward high-value regions, they often deviate from the optimal local direction, leaving nearby empty spaces that contain superior policies. Collectively, these observations suggest that in practice, on-policy RL struggles to provide accurate, low-variance gradient estimates, as single-batch updates remain heavily influenced by noise.

In on-policy algorithms (*e.g.*, PPO and TRPO), the sequence of iteration-end checkpoints drifts along these noisy directions, so the final checkpoint in each iteration can stray from high-reward regions, causing inefficiency and fluctuating performance. Merely increasing the batch size is not a viable solution, as even in simulated environments (e.g., MuJoCo), GPU memory and runtime constraints impose strict limits. Therefore, alternative mechanisms for correcting gradient drift, without incurring excessive data or computational costs, are essential for improving both efficiency and stability.

### 4.2 TRAINING GRANULARITY AND GRADIENT CORRECTION

**Policy training granularity** To outline where gradient correction can be applied in policy training, we define three levels of training granularity for on-policy RL: (i) iteration-level: the highest level, where fresh data are collected to populate the rollout buffer; (ii) epoch-level: an intermediate milestone within an iteration, after a certain number of parameter updates when a checkpoint is saved; (iii) batch-level: the lowest level, where a mini-batch is sampled from the rollout buffer to compute the gradient direction and update the parameters.

**Gradient correction method classification** According to our granularity framework, previous work mostly focuses at the batch-level correction that re-estimates gradients at every gradient calculation step; *e.g.*, Rahn et al. (2023) visualizes the policy return landscape at one gradient step ahead; Guided ES (Maheswaranathan et al., 2019) combines evolutionary strategies to search less biased gradient direction within one gradient update. However, each individual gradient step makes only a small parameter change, and performing repeated re-estimations over many updates incurs substantial data and computational overhead, yielding limited overall gains. Some recent work makes progress at the epoch level; *e.g.*, VFS (Marchesini & Amato, 2023) randomly searches for a better value network

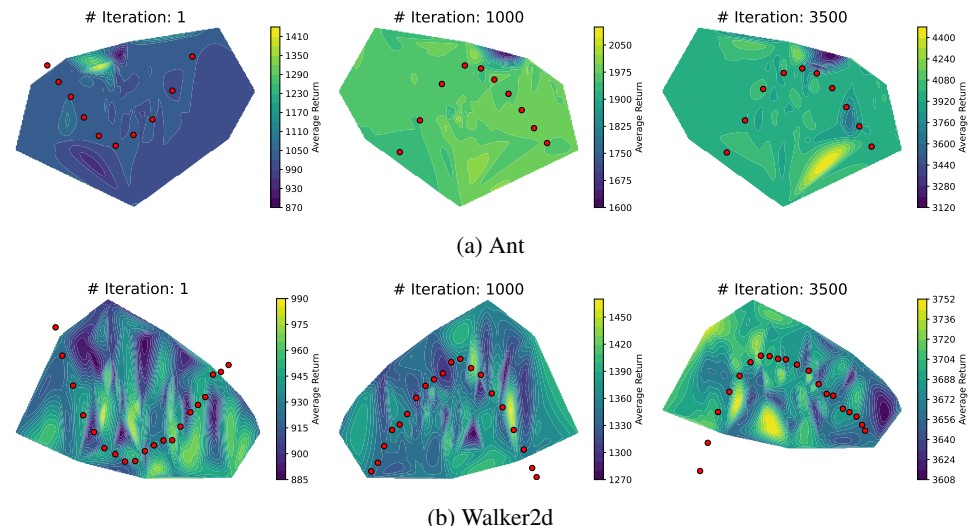

(a) Ant

(b) Walker2d

Figure 1: Visualization of policy value distribution in the local parameter space. Red dots denote 10 epoch checkpoints from a PPO iteration, fitted with a Gaussian to sample 100 candidate policies. Each candidate is evaluated over 1,000 episodes, and the average return is projected onto a PCA plane to form a contour map. Results are shown at iteration 1 (start), 1,000 (midpoint), and 3,500 (end). Figure 1a is the parameter space visualization for MuJoCo Ant and Figure 1b is for MuJoCo Walker2d. More results can be found in Appendix. A.

every multiple epochs to mitigate policy gradient error, but it searches around a single value network, missing the local information in the parameter space. These low-level methods focus on one-step correction or exploration around a single network. Once trapped in the local optima, they have to run a large amount of evaluations to correct the next-step gradient and escape the local optima step by step, which is less efficient. Iteration-level analysis leverages the structural information around checkpoints to explore the local parameter space more effectively, enabling faster escape from local optima. Recent work on iteration-level on-policy RL has primarily focused on algorithmic enhancements rather than gradient correction. For instance, Crowder et al. (2024) integrates HER (Andrychowicz et al., 2017) into the rollout buffer to adapt PPO for sparse-reward tasks. However, iteration-level gradient correction remains largely unexplored.

**Iteration-level gradient correction**  Our work, ExploRLer, performs a gradient correction at the iteration level. Thus, it reduces computational overhead in single updates and makes use of rich structural information in local parameter space. Figure 1 gives an intuitive illustration that within the local region defined by checkpoints, there exist "empty spaces" whose sampled policies yield higher returns than the checkpoints themselves. Relocating the policy to one of these high-return regions realizes an iteration-level correction. To implement this idea, we integrate ESA into the training loop. As a zero-order method, ESA can escape the surrogate-gradient direction and explore empty spaces around the checkpoints. The ESA proposals identify promising subregions as new starting points for following gradient estimations. Since ESA is independent of reinforcement learning, our method is agnostic to specific RL algorithms but serves as a pluggable component to enable gradient correction at each iteration during on-policy learning.

### 4.3  PARAMETER SPACE EXPLORATION PIPELINE

**Pipeline Details**  The full ExploRLer pipeline is described in Alg. 1. We empirically run ESA every 10 iterations in all our experiments (*i.e.*, $I = 10$). As mentioned earlier, the policy network can be trained by any on-policy learning algorithms. In this work, we instantiate and study two variants, ExploRLer + PPO (**ExploRLer-P**) and ExploRLer + TRPO (**ExploRLer-T**).

---

**Algorithm 1** ExploRLer Training Pipeline

---

**Require:** Initial policy $\pi_{\theta_1}$, online policy value evaluator $\hat{J}$, ESA interval $I$
 1: **for** iteration $i = 1$ to $K$ **do**
 2:    Collect rollouts using $\pi_{\theta_i}$
 3:    **for** epoch $e = 1$ to $n$ **do**
 4:       Update $\pi_{\theta_{i,e}}$ via on-policy learning
 5:    **end for**
 6:    Add the last checkpoint $\pi_{\theta_{i,n}}$ to the anchor set $\mathcal{A}_I$
 7:    **if** $i\%I == 0$ **then**
 8:       Apply ESA to $\mathcal{A}_I$ to collect $m$ candidate policies $\{\pi_{\psi_j}\}_{j=1}^m$
 9:       Set $\pi_{\theta_{i+1}} \leftarrow \arg\max_{\pi_{\psi_j}} \hat{J}(\pi_{\psi_j})$
10:       Clear $\mathcal{A}_I$
11:    **end if**
12: **end for**
**Ensure:** Final policy $\pi_{\theta_K}$

---

**ESA parameters**    ESA introduces additional parameters to the RL algorithms. We list the parameters below and explain the intuition behind the values. We set **(i) number of ESA agents** $m = n/2$ where $n$ is the number of epochs for on-policy RL; **(ii) number of agent neighbors** $N = 6$ to define local anchor geometry in the ESA graph. This was chosen to ensure stability in high-dimensional spaces where policy networks can diverge significantly (*e.g.*, in action distribution). We found empirically that smaller neighborhoods could lead to instability, while larger ones introduced excessive redundancy; **(iii) effective diameter** $\sigma$ to the average distance to 6 neighbors; **(iv) number of steps** $s = 60$ as agents tend to settle locally after 60 search steps; **(v) step size** $\alpha = 0.001$ to reflect typical learning rates in deep RL, enabling stable, fine-grained ESA search; **(vi) rollout interval** $I = 20$ to approximate the number of gradient updates per replay buffer in PPO (*e.g.*, 512-buffer size with 32-batch size gives 16 updates), aligning exploration granularity with on-policy learning's update rhythm. Note that we also roll out the initial position to our candidate pool. That said, we finally get $2n$ candidate policies for evaluation after ESA. Since fine-tuning parameters is expensive, we recommend the above default values for real scenario applications in all ExploRLer variants, and they prove to be effective in our experiments in Section 5.

**Online evaluation**    The online policy value evaluator $\hat{J}$ is defined by executing three full episodes per input candidate policy and outputs the average undiscounted returns. While an accurate average episode return of a policy requires thousands of episode evaluations, we reduce the evaluation number to three since a relative rank of the policies is sufficient. Empirically, three episode evaluations are enough to trade off the evaluation cost and the ranking accuracy. As shown in Section 5, this protocol reliably identifies high-performing candidates with minimal overhead.

## 5 EXPERIMENTAL RESULTS

We evaluate the effectiveness of ExploRLer-P and ExploRLer-T in on-policy RL through experiments conducted across several standard continuous control benchmarks. We aim to answer the following research questions: (a) How does ExploRLer perform compared to pure on-policy algorithms? (b) Can ExploRLer adapt to a wide range of environments? (c) Is the ESA operator effective in the ExploRLer pipeline? (d) How does ExploRLer perform compared to popular gradient correction methods?

We design two sets of experiments to answer the questions outlined above. The first set tests ExploRLer on low-variance environments including **Pendulum** from Classic Control and **BipedalWalker** from Box2D. They are simple environments that we use to verify the effectiveness of ExploRLer pipeline. The second set tests ExploRLer on continuous control environments from MuJoCo, including **(i) HalfCheetah** that has low-dimensional action and observation spaces and less variance in its environment; **(ii) Ant** that has higher action and observation dimensionality than HalfCheetah; **(iii) Hopper** that is sensitive to environment noise and highly variant in episode length, leading to a variant return distribution; **(iv) Walker2d** that is less variant compared to Hopper but has higher

action and observation dimensionality, resulting in a more complex policy space; **(v) Humanoid** that is high in both return distribution variance and observation and action space dimensionality.

Noticing the demand for data and computational resources for policy evaluation, we pretrain the policy network for 1 million gradient updates in all MuJoCo environments. Our ablation study in Section 5.3 shows that a pretrained policy does not drastically harm ExploRLer performance, but it mitigates the additional data and computational cost in the policy evaluation.

## 5.1 COMPARISON AGAINST ON-POLICY LEARNING

We answer questions (a) and (b) by comparing ExploRLer-P and ExploRLer-T with PPO and TRPO respectively, on the seven environments mentioned above. We train Pendulum and BipedalWalker for 200,000 and 1 million steps respectively; we train additional 2 million steps on top of pretrained policy for MuJoCo environments. The results are listed in Table 1. We can see that ExploRLer enhances the robustness of both PPO and TRPO, delivering higher returns across all the environments. One interesting finding is that although TRPO is worse than PPO on Walker2d, ExploRLer-T gets higher return than ExploRLer-P.

Table 1: Comparison of max average returns $\pm$ 1 std over 4 seeds.

|  | **PPO** | **ExploRLer-P** | **TRPO** | **ExploRLer-T** |
|---|---|---|---|---|
| Ant | $4433.69 \pm 71.03$ | $\mathbf{4573.98 \pm 190.18}$ | $3613.60 \pm 693.12$ | $3653.33 \pm 951.59$ |
| Hopper | $2233.97 \pm 934.81$ | $\mathbf{3318.46 \pm 82.62}$ | $556.31 \pm 75.97$ | $1060.49 \pm 640.28$ |
| Walker2d | $3647.18 \pm 506.37$ | $3762.48 \pm 467.38$ | $3263.77 \pm 575.37$ | $\mathbf{3778.41 \pm 515.20}$ |
| Humanoid | $547.26 \pm 121.76$ | $\mathbf{739.35 \pm 51.01}$ | $568.95 \pm 78.13$ | $686.96 \pm 83.87$ |
| HalfCheetah | $2237.39 \pm 1029.75$ | $2262.21 \pm 1162.43$ | $2589.41 \pm 1360.40$ | $\mathbf{2748.18 \pm 1359.39}$ |
| Pendulum | $-154.66 \pm 128.42$ | $\mathbf{-152.94 \pm 129.85}$ | $-222.28 \pm 97.21$ | $-222.55 \pm 96.57$ |
| BipedalWalker | $284.06 \pm 11.03$ | $\mathbf{293.17 \pm 6.81}$ | $-18.18 \pm 31.69$ | $174.65 \pm 117.09$ |

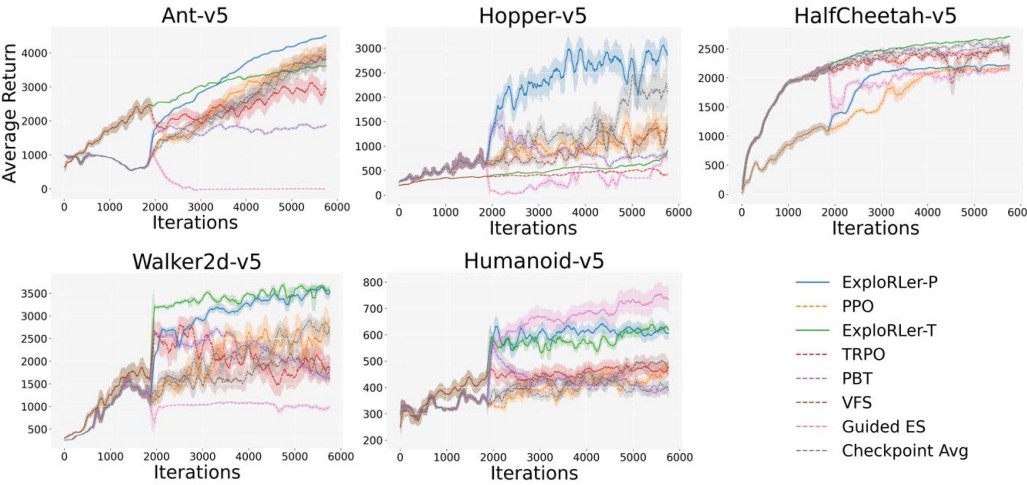

Figure 2: Comparison of training curves on MuJoCo environments across 3M steps. The solid and dashed lines show the average performance across 4 random seeds, with the shaded region indicating ±1 standard deviation, and with a smoothing window of length 100.

## 5.2 COMPARISON AGAINST BASELINES

To answer questions (c) and (d), we run a well-curated set of baselines on MuJoCo environments. To verify the effectiveness of ESA, we compare it with two intuitive methods that replaces the ESA operator with (1) Checkpoint averaging (Nikishin et al., 2018; Wortsman et al., 2022); (2) Population-based training (PBT) that selects the best half of the population, mutates and augments them, and finally selects the best one for the next iteration (Jaderberg et al., 2017; Li et al., 2024).

PBT also serves as an iteration-level baseline in our framework. Furthermore, we select Guided Evolutionary Strategies (Guided-ES) and Value Function Search (VFS) as additional baselines. They are popular gradient correction methods and serve as representative examples at the batch and epoch levels, respectively, within our training granularity framework. Please refer to Appendix D for details about hyperparameters, implementations, and hardware information.

The results are displayed in Figure 2. We first pretrain PPO and TRPO for 1 million steps in all environments before applying the gradient correction methods (the diverging points in the five plots). In the HalfCheetah environment, the baselines were inserted into TRPO, as it demonstrated superior performance to PPO. For all other environments, the baselines were inserted into PPO. Our proposed methods, ExploRLer-P and ExploRLer-T, were consistently inserted into PPO and TRPO, respectively, across all environments.

Figure 2 shows that ExploRLer consistently outperforms the baseline methods across most MuJoCo environments, with the only exception being Humanoid, where it ranks second. Interestingly, Guided-ES achieves competitive performance in Humanoid but fails to converge in other environments. This is likely due to the high level of noise in Humanoid, where batch-level gradient correction provides more robust updates. Nevertheless, even though Guided-ES performs better in Humanoid, ExploRLer still demonstrates significantly stronger and more stable performance than the other baselines overall.

### 5.3 ABLATION STUDY

We show the performance effect of pretraining policy for ExploRLer-P and ExploRLer-T in Figure 3. The results are mixed. None of the two strategies can dominate all environments. Therefore, we practically recommend pretraining the on-policy algorithm for certain steps (*e.g.*, 1 million steps) for three reasons: (i) Figure 3 proves that a pretrained policy saves data and computational cost for gradient correction without significantly harming the performance; (ii) some environments have a local optima trap at the early stage (*e.g.*, the unconditional survival reward in Ant) which is hard for zero-order methods to jump out, but policy gradient methods can capture the trap and escape; (iii) Ilyas et al. (2020) pointed it out that, in MuJoCo environments, policy gradient estimates align with true gradients early in training but diverge with the training progresses, so it's less necessary to do gradient correction at the early stage.

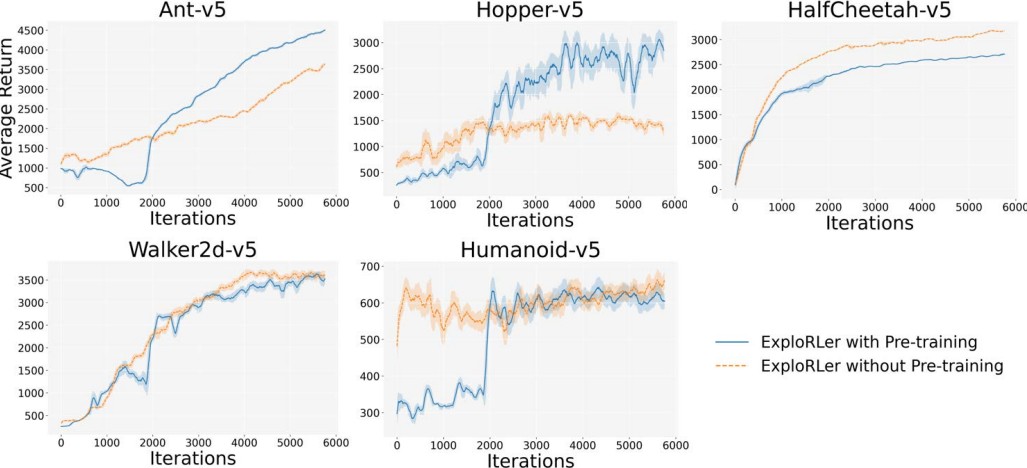

Figure 3: Training curves of Ablation Study for 3M steps. The solid and dashed lines show the average performance across 4 random seeds, with the shaded region indicating ±1 standard deviation, and with a smoothing window of length 100.

## 6 DISCUSSION, LIMITATION AND FUTURE WORK

**On-policy design rationale** We adopt an on-policy framework because it cleanly decouples the policy and value networks, facilitating modular integration of externally generated candidates. Off-

policy methods such as SAC rely on tightly coupled Q networks, target networks, and stochastic policies, complicating the insertion of new policy parameters without introducing large approximation errors. Figure 4 provides an example illustrating the limited effectiveness of ESA when applied to SAC. In contrast, on-policy algorithms estimate gradients directly from the policy and train the value function separately for advantage estimation. This separation allows us to safely replace the policy network via ESA proposals and online evaluation without destabilizing other components. Applying gradient correction from parameter space exploration to off-policy learning is a promising avenue for future work.

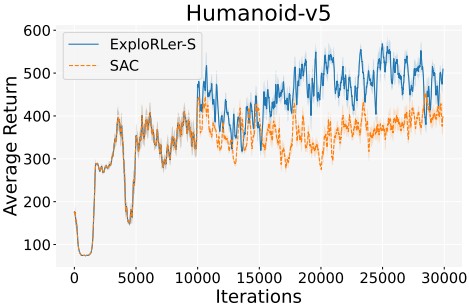

Figure 4: Training curves for Humanoid-v5 using SAC with and without ExploRLer integration. The solid and dashed lines show the average performance across 4 random seeds, with shaded regions indicating ±1 standard deviation, and with a smoothing window of length 100.

**Policy evaluation efficiency** One limitation of our work is the extra sample and computational overhead in policy evaluations required by ESA. Specifically, ESA is invoked every 10 training iterations, generating 20 candidate policies that are each evaluated over 3 rollout episodes, adding 6 extra episodes per iteration on average. Appendix C lists the runtime cost of all methods we test. In addition to online evaluation, alternative solutions have been sufficiently discussed in gradient correction research, for example, the world model (Ha & Schmidhuber, 2018) and off-policy evaluation (OPE) (Zhang et al., 2022). In simulation-based environments, a feasible way to accelerate the training is to run policy evaluation in parallel as candidate policies can be evaluated independently. Figure 5 compares policy evaluation strategies by integrating Fitted Q Evaluation (FQE, an OPE method) with online evaluation in ExploRLer. The hybrid approach reduces the number of online interactions while retaining competitive performance, suggesting that FQE can serve as an effective pre-filtering mechanism. However, fully replacing online evaluation with OPE leads to unreliable candidate selection, confirming that standalone OPE under non-stationary data remains an open challenge.

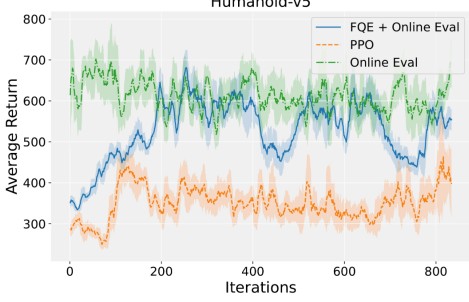

Figure 5: The orange line is the regular PPO algorithm; the green line is ExploRLer-P with fully online evaluation; the blue line is ExploRLer with FQE and online evaluation combination. All the experiments start from a 1-million-step pretrained model.

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

## A    PARAMETER SPACE VISUALIZATIONS

To complement the visualizations in Figure 1, Figure  6 presents additional local parameter space results for Hopper, Humanoid, and HalfCheetah. The visualizations demonstrate that, consistent with the results for Ant and Walker2d, checkpoints frequently diverge from adjacent high-value regions, thereby creating empty spaces that contain superior candidate policies. The systematic occurrence of this phenomenon across diverse environments provides strong empirical support for adopting iteration-level exploration in ExploRLer.

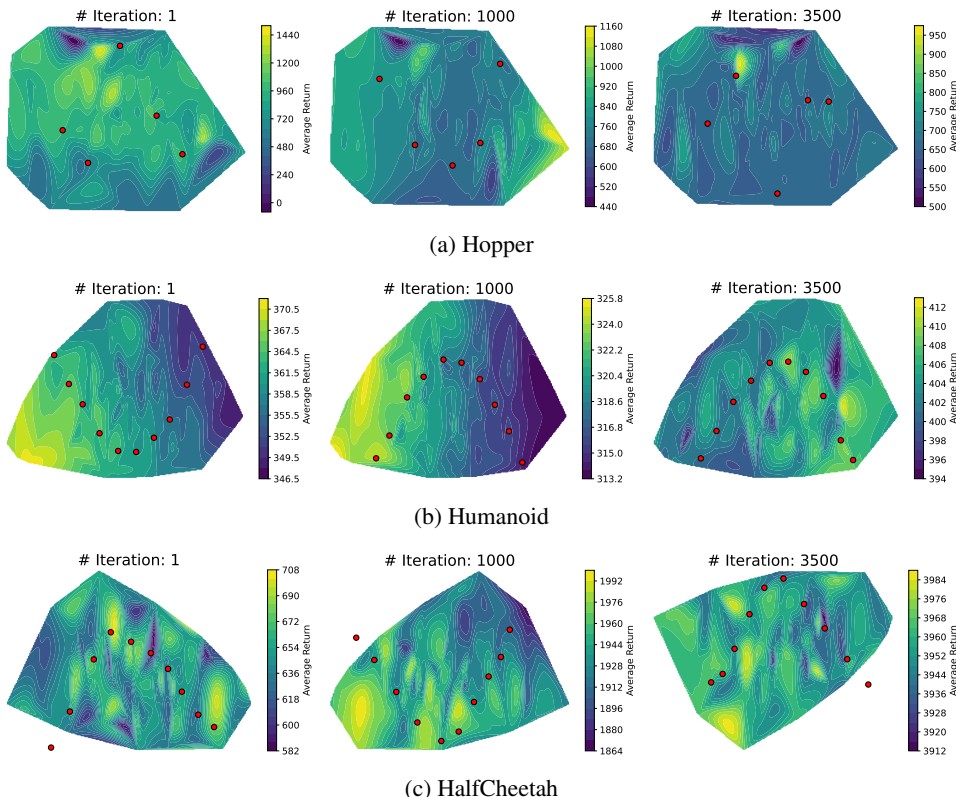

(a) Hopper

(b) Humanoid

(c) HalfCheetah

Figure 6: Local parameter space visualizations for Hopper, Humanoid, and HalfCheetah, showing the distribution of checkpoints within the parameter space and revealing adjacent empty regions that can host higher-value candidate policies.

## B    FURTHER EXPERIMENTS ON CLASSIC CONTROL AND BOX2D

We train Pendulum and BipedalWalker for 200,000 and 1 million steps respectively, using 4 random seeds per method. Fig 7 illustrates the performance of ExploRLer compared to the base on-policy algorithms on both environments. On BipedalWalker, ExploRLer achieves a clear improvement in average return compared to the conventional baselines. For Pendulum, ExploRLer exhibits faster early-stage learning and surpasses the baseline performance during the initial training phase.

## C    COMPUTATIONAL COST

The impact on runtime is summarized in Table 2. PPO iterations that typically complete within a few seconds increase substantially when augmented with ExploRLer. This overhead reflects the additional evaluations introduced by ESA and represents a clear limitation of the approach. However, despite higher per-iteration costs, ExploRLer often achieves stronger per-step performance improvements, which can offset the added runtime when considering overall training efficiency.

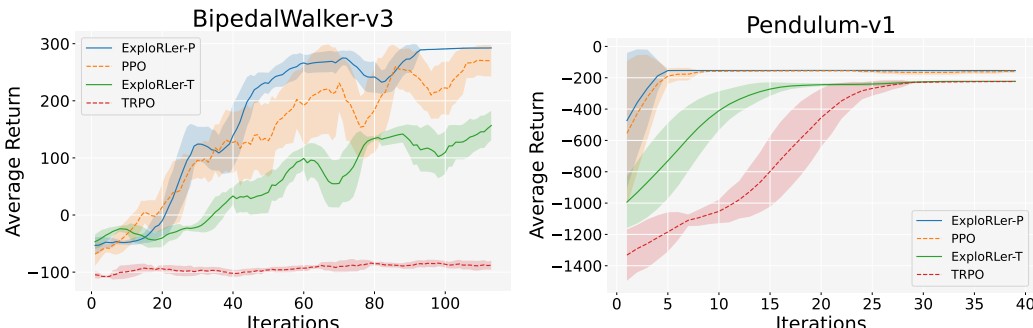

Figure 7: Training curves for Classic Control and Box2D environments. The solid and dashed lines show the average performance across 4 random seeds, with the shaded region indicating ±1 standard deviation, and with a smoothing window of length 10.

Table 2: Comparison of average wall-clock time per iteration (in seconds) across environments.

|  | Exp-P | Exp-T | PPO | TRPO | PBT | VFS | Guided-ES | Checkpoint Avg |
|---|---|---|---|---|---|---|---|---|
| Ant | 75.48 | 80.62 | 3.96 | 3.75 | 50.96 | 4.29 | **99.27** | 3.48 |
| HalfCheetah | 48.54 | 54.57 | 2.46 | 11.13 | 41.94 | 11.85 | **80.62** | 11.09 |
| Hopper | **121.11** | 18.99 | 1.67 | 10.10 | 21.49 | 2.94 | 18.53 | 2.96 |
| Walker2d | 39.27 | 34.48 | 3.32 | 1.97 | 28.84 | 3.12 | **95.42** | 2.56 |
| Humanoid | **48.82** | 33.58 | 1.90 | 1.36 | 24.45 | 4.09 | 46.40 | 3.77 |

# D    IMPLEMENTATION DETAILS OF EXPLORLER

**Network Architecture**    We use the Proximal Policy Optimization (PPO) and Trust Region Policy Optimization (TRPO) algorithm as implemented in Stable Baselines3 (Raffin et al., 2021) as the base learners in our pipeline. For both algorithms, the policy and value functions are parameterized by a shared multilayer perceptron (MLP) with two hidden layers of 64 units each and ReLU activations. The policy head outputs action distributions, while the value head estimates state values.

PPO optimizes the actor and critic jointly using the clipped surrogate objective and value function loss, combined with an entropy bonus to encourage exploration. TRPO instead employs a constrained optimization approach, enforcing a trust region based on the KL-divergence between successive policies to guarantee stable improvement. Both implementations include standard techniques such as generalized advantage estimation (GAE), gradient clipping, and learning rate annealing by default. Unless otherwise noted, we retain the default architectural and training configurations from Stable Baselines3.

**ESA Implementation**    We train the base on-policy algorithm and run Empty-Space Search (ESA) every 10 iterations. Each iteration has multiple epochs. We save the last epoch checkpoint from each iteration as the anchor point for the ESA. The candidate policies identified by ESA are evaluated over three online episodes, and ranked based on their average returns. The top-performing policy is then loaded back into the model, and this cycle is repeated throughout training.

**Hyperparameters for RL Algorithms**    Table 3 and Table 4 presents the detailed hyperparameters of PPO and TRPO trained on different tasks. We utilize the pre-tuned optimal hyperparameters provided by RL Baselines3 Zoo (Raffin, 2020) for all baseline implementations, ensuring standardized comparison across environments and algorithms.

**Hardware**    We train our models on CPU with the Intel Xeon Gold 6140 CPU (each with 2.30GHz, 36 cores, 25MB Cache) and 1.5TB DDR4 RAM. Pre-training typically takes around 1–2 hours, while the full training process requires an additional 4–5 hours, depending on the complexity of the environment.

Table 3: Hyperparameters of PPO used for different tasks

| Environments | Learning Rate | Clip Range | Steps per rollout | No. of Envs | Batch Size | Discount Factor | Entropy Coefficient | GAE lambda |
|---|---|---|---|---|---|---|---|---|
| Ant | $1.90609e^{-05}$ | 0.1 | 512 | 1 | 32 | 0.98 | $4.9646e^{-07}$ | 0.8 |
| Humanoid | $1.90609e^{-05}$ | 0.1 | 512 | 1 | 32 | 0.98 | $4.9646e^{-07}$ | 0.8 |
| Walker2d | $5.05041e^{-05}$ | 0.1 | 512 | 1 | 32 | 0.99 | $5.85045e^{-04}$ | 0.95 |
| HalfCheetah | $1.90609e^{-05}$ | 0.1 | 512 | 1 | 32 | 0.98 | $4.9646e^{-07}$ | 0.8 |
| Hopper | $9.80800e^{-05}$ | 0.2 | 512 | 1 | 32 | 0.99 | $2.29500e^{-03}$ | 0.99 |
| BipedalWalker | $3e^{-04}$ | 0.18 | 2048 | 4 | 64 | 0.99 | 0.0 | 0.95 |
| Pendulum | $1e^{-03}$ | 0.2 | 1024 | 4 | 64 | 0.9 | 0.0 | 0.95 |

Table 4: Hyperparameters of TRPO used for different tasks

| Environments | Learning Rate | CG Max Steps | Steps per rollout | No. of Envs | Batch Size | Discount Factor | Critic Updates | GAE lambda |
|---|---|---|---|---|---|---|---|---|
| Ant | $1.90609e^{-05}$ | 25 | 512 | 1 | 32 | 0.98 | 20 | 0.8 |
| Humanoid | $1.90609e^{-05}$ | 25 | 512 | 1 | 32 | 0.98 | 20 | 0.8 |
| Walker2d | $5.05041e^{-05}$ | 25 | 512 | 1 | 32 | 0.99 | 20 | 0.95 |
| HalfCheetah | $1.90609e^{-05}$ | 25 | 512 | 1 | 32 | 0.98 | 20 | 0.8 |
| Hopper | $1.90609e^{-05}$ | 25 | 512 | 1 | 32 | 0.98 | 20 | 0.8 |
| BipedalWalker | $1.90609e^{-05}$ | 25 | 512 | 1 | 32 | 0.98 | 20 | 0.8 |
| Pendulum | $1e^{-03}$ | 15 | 1024 | 2 | 128 | 0.9 | 15 | 0.95 |

# E  IMPLEMENTATION DETAILS OF BASELINES

To ensure fair comparisons, all baselines were adapted to operate at the iteration-level, mirroring how ESA is applied in our framework. We briefly describe each baseline below:

**Checkpoint Averaging**  We average the parameters of all epoch checkpoints within one iteration to obtain a candidate policy. This averaged policy replaces the ESA-generated candidate before evaluation. All other training settings remain identical to ExploRLer-P and ExploRLer-T.

**Population-Based Training (PBT)**  At each iteration, we maintain a population of 10 policy checkpoints. Each policy is evaluated over three online episodes to obtain returns. The top five policies are retained, while the bottom five are replaced with mutated copies of the top set. Mutation is performed by adding Gaussian noise ($\sigma = 0.02$) to the parameters of a randomly chosen top-performing policy. This ensures that high-performing policies are preserved while lower-performing ones are replaced with promising variants, keeping the population size fixed.

**Guided Evolutionary Strategies (Guided-ES)**  We implement Guided ES by combining surrogate gradients from PPO with zeroth-order evolutionary estimates. At each iteration, we flatten the policy parameters (excluding the value network) and compute the current PPO gradient through a dummy backward pass. A set of symmetric perturbations ($\theta \pm \sigma\epsilon$, with $\sigma = 0.02$) is then sampled, and each perturbed policy is evaluated over three episodes to obtain an ES gradient estimate. The final update direction is formed as a convex combination of the PPO gradient and the mean ES gradient, with mixing coefficient $\alpha = 0.5$. This combined update is used to construct a new candidate policy, which is then inserted back into training.

**Value Function Search (VFS)**  We adapt the original VFS idea for our on-policy setting. Instead of perturbing the value network directly, we use the value estimate to guide small updates to the policy parameters. Concretely, at each iteration we sample an observation, backpropagate through the value head, and update only the policy parameters (excluding the value network) with gradient ascent steps of size $\alpha = 0.01$. We perform $k = 3$ such steps and record the resulting updated policy as a candidate. The original policy parameters are restored after this procedure to avoid interfering with the main training loop.

For all baselines, we used the same network architecture (two-layer MLP with 64 hidden units, ReLU activations) and optimizer settings as PPO/TRPO to ensure comparability. Training was conducted under the same hardware and random seed protocols as ExploRLer.

## F ABLATION STUDY ON EMPTY-SPACE SEARCH

We perform an additional ablation study to assess the importance of the Empty-Space Search (ESA) module in discovering high-quality agents. To do this, we replace ESA with a random walk strategy and evaluate performance across various tasks. The results in Fig. 8 show that the Empty-Space Search (ESA) consistently outperforms the random walk strategy across most environments. ESA not only accelerates learning but also achieves higher final returns, highlighting its effectiveness in guiding exploration.

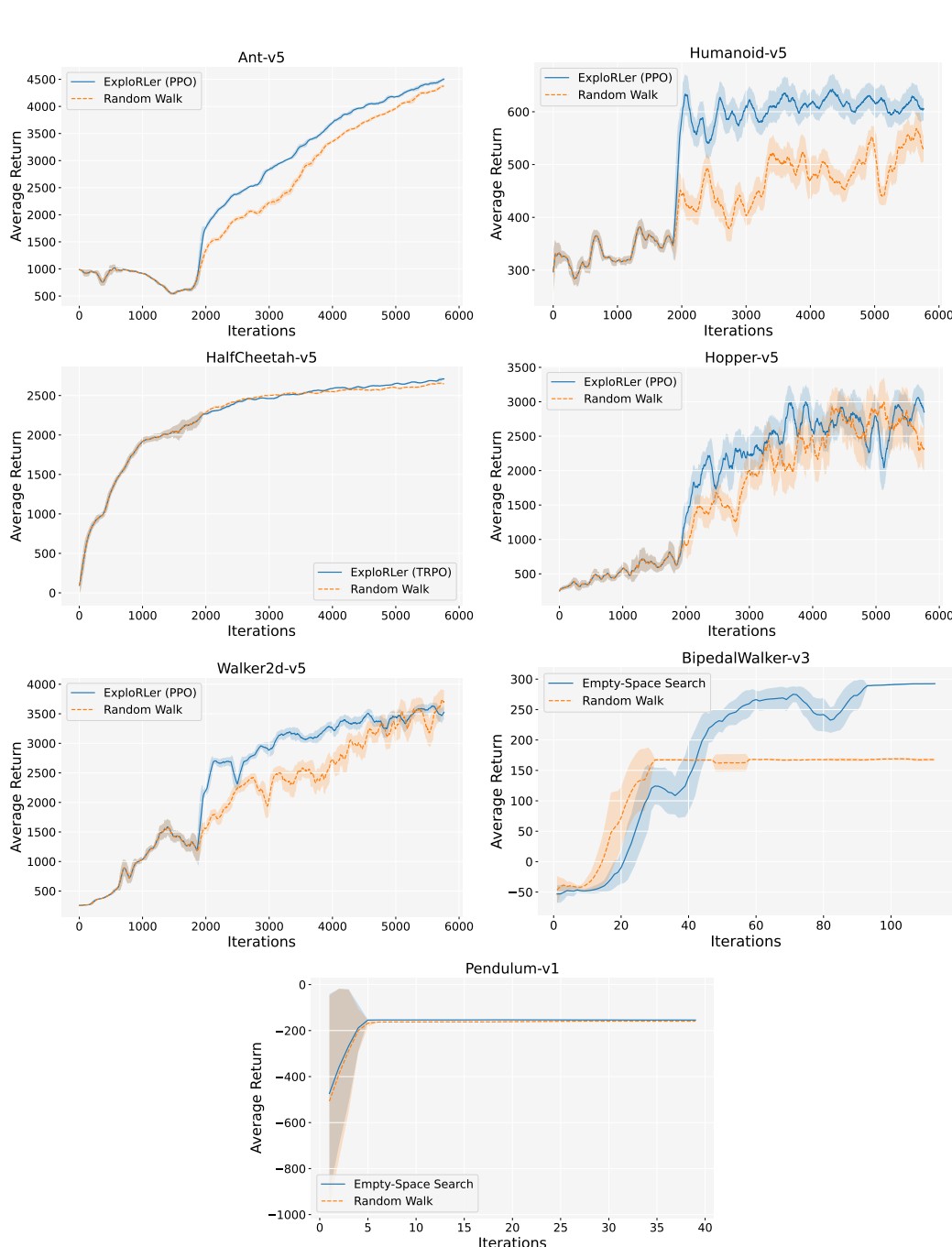

Figure 8: Training curves of ExploRLer for different agent search techniques. The solid and dashed lines show the average performance across 4 random seeds, with the shaded region indicating ±1 standard deviation.

