# OpenReview forum: "Efficient On-Policy Reinforcement Learning via Exploration of Sparse Parameter Space"
_ICLR.cc/2026/Conference — Submitted to ICLR 2026_

### Official Review · Reviewer_F9pR · 2025-10-25

**Soundness:** 2
**Presentation:** 2
**Contribution:** 1
**Rating:** 2
**Confidence:** 5

**Summary:**

This paper proposes a method named ExploRLer that probes the unexplored neighborhoods of surrogate on-policy gradient updates.

**Strengths:**

A straight-forward paper motivated from an interesting problem.

**Weaknesses:**

+ **Critical information is missing:** The paper repeatedly mentions ESA, an algorithm in the previous work which serves as the keystone of the proposed method. However, the ESA algorithm is never clearly introduced and I still have no idea about what it really is after reading the paper. The introduction in Section 3.2 does not give me enough knowledge to fully understand it. Therefore, I am unable to evaluate the soundness and effectiveness of the proposed ExploRLer method.

+ **Baselines are not well-tuned:** For example, it has been shown that a well-tuned SAC algorithm can achieve a cumulative reward around 6000 on the Humanoid task (e.g., https://arxiv.org/pdf/1812.05905, Figure 1), whereas it merely reaches 400 as in Figure 4 in this paper. Also, TRPO should achieve at least 2000 on the Hopper (https://arxiv.org/pdf/1707.06347), but in Table 1 it is only 556.31.

+ **Limited contribution:** The main part of this paper, saying Section 4, is just a recap of existing work and the algorithm design, without giving me any insight why the proposed algorithm would work. Also, it is difficult for me to understand what Figure 1 means. It is likely due to the insufficient information I have about empty spaces and ESA, but the authors should not assume that the audience have all prerequisite knowledge.

Overall, the paper needs to be re-organized and I hence recommend reject.

**Questions:**

I have no question.

---

### Official Review · Reviewer_p8VC · 2025-10-28

**Soundness:** 2
**Presentation:** 3
**Contribution:** 3
**Rating:** 4
**Confidence:** 4

**Summary:**

The paper proposes ExploRLer, a plug-and-play augmentation for on-policy RL (PPO/TRPO) that performs iteration-level parameter-space exploration via an Empty-Space Search (ESA) operator. After every I iterations, the method collects recent checkpoints as anchors, samples candidate policies in their “empty” neighborhoods, evaluates each candidate with a small number of online rollouts, and resumes training from the best candidate. The goal is to counter surrogate-gradient drift and exploit local structure missed by a single stochastic gradient path. Contributions claimed: (i) the ExploRLer pipeline; (ii) a training-granularity framing (batch/epoch/iteration) motivating iteration-level corrections; and (iii) empirical gains over PPO/TRPO and several baselines on MuJoCo, Box2D, and Classic Control.

**Strengths:**

1.The paper has a clear and simple integration: ESA is inserted at iteration boundaries without extra per-mini-batch gradient overhead; the algorithmic scaffold (Algorithm 1) is straightforward to reimplement.

2.The paper is motivated by an observed failure mode (surrogate-gradient drift) with compelling local landscape visualizations that match intuition from prior analyses.

3.Results across MuJoCo, Box2D, Classic Control; table shows consistent gains in many tasks (often higher final returns and sometimes faster convergence).

**Weaknesses:**

1.Algorithmic details are specified (Algorithm 1, ESA hyperparameters, evaluation protocol), but important design choices appear under-justified (e.g., evaluation with only 3 episodes per candidate; fixed ESA interval and agent counts; choice of neighbors; step size), and their sensitivity is not systematically probed.

2.Using 3 online episodes per candidate to pick the next starting point is fragile in high-variance tasks (e.g., Humanoid). The paper claims this is “enough” for ranking but does not provide calibration analyses.

3.Most MuJoCo experiments start from a 1M-step pretrained policy. While the paper argues this reduces cost and doesn’t “drastically harm” performance, it leaves unclear whether ExploRLer is beneficial from scratch under equal wall-clock or interaction budgets. A clean comparison without pretraining (or matched total interactions) is necessary to support sample-efficiency claims.

**Questions:**

See Weaknesses

---

### Official Review · Reviewer_At5A · 2025-10-31

**Soundness:** 2
**Presentation:** 2
**Contribution:** 2
**Rating:** 2
**Confidence:** 5

**Summary:**

The paper proposes ExploRLer, a reinforcement learning (RL) framework that combines on-policy RL algorithms such as PPO and TRPO with empty-space search in policy parameter space. The method stores policy checkpoints after every iteration, and periodically uses the most recent checkpoints as the starting point for an empty-space search to find improved policy parameters. Experiments compare performance of ExploRLer to on-policy baselines and other methods that incorporate search across 7 benchmark tasks (Pendulum, BipedalWalker, Ant, Hopper, HalfCheetah, Walker2d, Humanoid).

**Strengths:**

**[S1]** The paper proposes a novel application of empty-space search in RL training.

**[S2]** On-policy methods such as PPO have become a popular solution for real-world applications in robotics and foundation models, and the proposed algorithm builds upon these popular methods.

**Weaknesses:**

**[W1]** The paper does not provide any theoretical analysis or guarantees related to the proposed algorithm.

**[W2]** The proposed method introduces several new hyperparameters that need to be tuned (and are not clearly discussed or analyzed), requires a potentially significant increase in memory for large policy networks (it stores the last 10 policy checkpoints and performs a search in policy parameter space), and adds meaningful computational and sample cost to evaluate candidate policies (often ~20x slower than PPO according to Table 2).

**[W3]** The paper claims that “increasing the batch size is not a viable solution” (lines 196-197), but in recent years highly parallelized implementations of PPO with very large batch sizes have led to strong performance on real-world robotics tasks such as legged locomotion [1]. The paper argues that large batch sizes may lead to memory or runtime issues, but the proposed method also introduces additional memory and computational requirements.

**[W4]** Experimental results are limited and do not provide convincing support for the proposed algorithm given the added complexity, memory, and computation it requires.
- Experiments consider very small networks (2 hidden layers with 64 units) and very small batch sizes (512), leading to sub-par performance of baselines relative to results available in the literature.
- Experiments only consider a small set of basic benchmark tasks. Given the lack of theoretical analysis, I would expect the experimental analysis to demonstrate convincing performance benefits across a wide range of realistic tasks, including realistic robotics tasks (e.g., legged locomotion, manipulation, etc. in Isaac Lab [2]) and/or LLM applications as mentioned in the Introduction (lines 29-30).
- ExploRLer should also be compared to parallelized on-policy methods with large batch sizes. Because ExploRLer does not have guarantees and requires additional memory and computation beyond on-policy methods, it would also be reasonable to compare ExploRLer against off-policy methods that have been shown to achieve strong performance on these benchmark tasks.

**References:**

[1] Rudin et al., “Learning to walk in minutes using massively parallel deep reinforcement learning.” In CoRL 2021.

[2] Mittal et al., “Orbit: A unified simulation framework for interactive robot learning environments.” IEEE RA-L, 2023. https://github.com/isaac-sim/IsaacLab

**Questions:**

**[Q1]** Please compare against well-tuned versions of all baselines. While I understand the hyperparameters come from an external source, they do not reflect common choices for this set of MuJoCo benchmark tasks. The current analysis also only compares performance across number of iterations, but ExploRLer requires additional data and computation time for each iteration compared to the on-policy baselines. It would be more meaningful to compare ExploRLer to baselines (on-policy, on-policy with large batch size, off-policy) for the same amount of total wall-clock training time.

**[Q2]** Does ExploRLer improve the performance of on-policy methods such as PPO when large batch sizes are used? Large batch sizes have become very common when applying PPO to complex robotics tasks such as legged locomotion [1].

**[Q3]** How does ExploRLer scale when applied with larger policy networks which are common in more complex tasks, given empty-space search occurs in policy parameter space and requires multiple checkpoints to be stored in memory?

**[Q4]** How does resetting the policy based on the ESA impact the value function estimate? The distribution shift induced by larger policy changes could lead to additional error in the value function, and the use of a shared MLP as described in Appendix D would also cause error in the value function when resetting policy parameters (because the policy and value share these parameters).

---

### Official Review · Reviewer_bARV · 2025-11-01

**Soundness:** 2
**Presentation:** 2
**Contribution:** 1
**Rating:** 2
**Confidence:** 4

**Summary:**

This paper proposes ExploRLer, a parameter-space exploration algorithm on-policy RL. The method runs an Empty-Space Search (ESA) procedure in the local parameter region to propose candidate policies, evaluates candidates online, and then resumes training from the best one. A core benefit of this approach is that it requires no additional gradient computation. Experiments show improvements over vanilla PPO and TRPO in standard MuJoCo tasks.

**Strengths:**

1. ExploRLer is compatible with any on-policy RL algorithm (and off-policy algorithms as well, I think).
2. Experiments potentially show improvements over vanilla baselines.
3. Appendices provide useful ablation and reproducibility details.

**Weaknesses:**

**I vote to reject primarily due to (1) inadequate baselines and (2) lack of clarity about the method’s motivation, algorithm, and contributions.**

1. **Inadequate Baselines.** Experiments only compare PPO/TRPO with and without ExploRLer, but numerous on-policy exploration methods already exist: intrinsic motivation, count-based exploration, Thompson sampling, even parameter-space exploration [1–8]. None of these are cited or included as baselines, which makes it difficult to evaluate whether ExploRLer provides any meaningful improvement beyond established techniques. If excluding these baselines is intentional, the paper should clearly justify this choice and, at minimum, include a discussion of these methods in the related-work section.

    Moreover, ExploRLer evaluates roughly 20 candidates over 3 episodes every I iterations, so although the authors emphasize “zero additional gradient computation per mini-batch,” the method collects substantially more trajectories than the vanilla baselines. To assess fairness, we need to know whether ExploRLer outperforms simply increasing the batch size by the same factor (and scaling the mini-batch size by the same factor to ensure that both methods perform the same number of gradient updates).

2. **Statistical Significance.** Results are averaged over five seeds, which is too few to draw reliable conclusions. Prior work (Henderson et al., AAAI 2018) shows that even identical RL algorithms can appear statistically distinct when evaluated on a small number of trails (like n=5).. Given the high variance of continuous-control benchmarks, 5 seeds do not provide convincing evidence of improvement.

3. **Algorithm Motivation and Clarity.** While the motivation for exploration in on-policy RL is sound, it remains unclear why ExploRLer’s Empty-Space Search (ESA) is preferable to existing exploration strategies. ESA itself is only briefly referenced and not explained; readers unfamiliar with the prior work cannot easily understand or reproduce the method.

# My suggestions
* Report total environment interactions (including evaluation episodes) and plot return vs. total steps and return vs. wall-clock time.
* Include a larger-batch PPO/TRPO baseline or normalize results by total interaction cost.
* Add a subset of the most relevant existing parameter-space exploration methods as baselines.
* Run at least 20–50 seeds, or if computationally infeasible, follow Agarwal et al., 2021 and report aggregate normalized performance across environments rather than individual learning curves. Provide 95% bootstrap confidence intervals to support improvement claims.
* Clearly summarize how ESA works, at least at a high level, so the paper is self-contained. Articulate what makes ESA distinct from existing parameter-space exploration techniques.

---

# References
1. Pathak et al., Curiosity-Driven Exploration by Self-Supervised Prediction.
https://arxiv.org/abs/1705.05363
1. Sukhbaatar et al., Intrinsic Motivation and Automatic Curricula via Asymmetric Self-Play.
https://arxiv.org/abs/1703.05407
1. Tang et al., #Exploration: A Study of Count-Based Exploration for Deep Reinforcement Learning.
https://arxiv.org/abs/1611.04717
1. Ostrovski et al., Count-Based Exploration with Neural Density Models..
https://arxiv.org/abs/1703.01310
1. Osband et al., Deep Exploration via Bootstrapped DQN.
https://arxiv.org/abs/1602.04621
1. Sutton and Barto, Reinforcement Learning: An Introduction, 2nd ed., 2018.
https://arxiv.org/abs/1805.00909
1. Plappert et al., Parameter Space Noise for Exploration.
https://arxiv.org/abs/1706.01905
1. On-Policy Policy Gradient Reinforcement Learning Without On-Policy Sampling.
https://arxiv.org/abs/2311.08290

**Questions:**

1. How sensitive is performance to I (ESA frequency) and m (candidate count)? Can fewer candidates (e.g., m/2) preserve most gains, reducing eval overhead? Do selection results change if you double the per-candidate episodes to 6 but halve the candidate count—same eval budget, different bias/variance trade-off?
2. In table 1, does "max average returns" refer to the maximum point on each training curve?
3. I don't understand the decision to pre-train the policy. Why was this done?

---

### Meta-Review · Area_Chair_b56j · 2025-12-30

**Summary:**

1.Lack of many exploration-based baselines, proposed by Reviewer bARV.

2.The motivation why ExploRLer work is not clear,  proposed by Reviewer bARV.

3.Lack of larger or realistic baselines, proposed by Reviewer At5A.

4.A clean comparison without pretraining is necessary to support sample-efficiency claims, proposed by by Reviewer p8VC.

5.Baselines such SAC are not well-tuned, proposed by Reviewer F9pR.

**Reviewer Concerns:**

The authors does not submit the rebuttal.

**Reviewer Scores:**

The authors does not submit the rebuttal, and the reviewers would not changed their score if they had been able to participate fully in the discussion.

---

### Decision · Program_Chairs · 2026-01-26

Reject